# Robust Saliva-Based RNA Extraction-Free One-Step Nucleic Acid Amplification Test for Mass SARS-CoV-2 Monitoring

**DOI:** 10.3390/molecules26216617

**Published:** 2021-10-31

**Authors:** Eva Rajh, Tina Šket, Arne Praznik, Petra Sušjan, Alenka Šmid, Dunja Urbančič, Irena Mlinarič-Raščan, Polona Kogovšek, Tina Demšar, Mojca Milavec, Katarina Prosenc Trilar, Žiga Jensterle, Mihaela Zidarn, Viktorija Tomič, Gabriele Turel, Tatjana Lejko-Zupanc, Roman Jerala, Mojca Benčina

**Affiliations:** 1Department of Synthetic Biology and Immunology, National Institute of Chemistry, SI-1001 Ljubljana, Slovenia; eva.rajh@ki.si (E.R.); tina.sket@ki.si (T.Š.); arne.praznik@ki.si (A.P.); petra.susjan@ki.si (P.S.); roman.jerala@ki.si (R.J.); 2Faculty of Pharmacy, University of Ljubljana, SI-1000 Ljubljana, Slovenia; alenka.smid@ffa.uni-lj.si (A.Š.); dunja.urbancic@ffa.uni-lj.si (D.U.); Irena.Mlinaric@ffa.uni-lj.si (I.M.-R.); 3Department of Biotechnology and Systems Biology, National Institute of Biology, SI-1000 Ljubljana, Slovenia; polona.kogovsek@nib.si (P.K.); tina.demsar@nib.si (T.D.); Mojca.Milavec@nib.si (M.M.); 4National Laboratory of Health, Environment, and Food, Laboratory for Public Health Virology, SI-1000 Ljubljana, Slovenia; katarina.prosenc.trilar@nlzoh.si (K.P.T.); Ziga.Jensterle@hnmp.info (Ž.J.); 5Emergency Service, Health Centre Jesenice, SI-4270 Jesenice, Slovenia; Mihaela.Zidarn@klinika-golnik.si; 6University Clinic of Respiratory and Allergic Diseases, SI-4204 Golnik, Slovenia; Viktorija.Tomic@klinika-golnik.si; 7Department for Infectious Diseases, University Medical Center Ljubljana, SI-1000 Ljubljana, Slovenia; gabriele.turel@gmail.com (G.T.); tatjana.lejko@kclj.si (T.L.-Z.); 8EN-FIST Centre of Excellence, SI-1000 Ljubljana, Slovenia; 9Biotechnical Faculty, University of Ljubljana, SI-1000 Ljubljana, Slovenia

**Keywords:** saliva, COVID-19, SARS-CoV-2, LAMP, RT-qPCR, passive drool, oral cavity swab, pooling

## Abstract

Early diagnosis with rapid detection of the virus plays a key role in preventing the spread of infection and in treating patients effectively. In order to address the need for a straightforward detection of SARS-CoV-2 infection and assessment of viral spread, we developed rapid, sensitive, extraction-free one-step reverse transcription-quantitative polymerase chain reaction (RT-qPCR) and reverse transcription loop-mediated isothermal amplification (RT-LAMP) tests for detecting SARS-CoV-2 in saliva. We analyzed over 700 matched pairs of saliva and nasopharyngeal swab (NSB) specimens from asymptomatic and symptomatic individuals. Saliva, as either an oral cavity swab or passive drool, was collected in an RNA stabilization buffer. The stabilized saliva specimens were heat-treated and directly analyzed without RNA extraction. The diagnostic sensitivity of saliva-based RT-qPCR was at least 95% in individuals with subclinical infection and outperformed RT-LAMP, which had at least 70% sensitivity when compared to NSBs analyzed with a clinical RT-qPCR test. The diagnostic sensitivity for passive drool saliva was higher than that of oral cavity swab specimens (95% and 87%, respectively). A rapid, sensitive one-step extraction-free RT-qPCR test for detecting SARS-CoV-2 in passive drool saliva is operationally simple and can be easily implemented using existing testing sites, thus allowing high-throughput, rapid, and repeated testing of large populations. Furthermore, saliva testing is adequate to detect individuals in an asymptomatic screening program and can help improve voluntary screening compliance for those individuals averse to various forms of nasal collections.

## 1. Introduction

The COVID-19 pandemic caused by the SARS-CoV-2 virus has, since its beginning in 2020, caused a worldwide humanitarian crisis, with an unprecedented death toll and widespread economic crises (https://reliefweb.int/topics/covid-19-global, accessed on 10 October 2020). In addition to vaccination, early detection, isolation, and management of infected individuals all play critical roles in avoiding further escalation of the pandemic. The basic tools for preventing the spread of the virus among the population are disease screening, isolation of infected individuals, identifying and alerting asymptomatic individuals, and contact tracing. 

Viral diagnosis has tremendously progressed with many recent and accurate techniques, ranging from laboratory nucleic acid amplification testing to point-of-care (PoC) testing. A nasopharyngeal swab (NPS) assessed by reverse transcription-quantitative polymerase chain reaction (RT-qPCR) is the recommended diagnosis method for COVID-19 detection [1,2]. However, NPS sampling is more challenging than collecting certain other types of biological samples (e.g., saliva). First, for NPS collection, trained personnel are required; this exposes the staff to a higher risk of infection, is costly, and therefore less suitable for mass testing. Furthermore, NPS sampling causes discomfort to individuals and is associated with contraindications, such as coagulopathy, or anticoagulant therapy, and nasal septum deviations. 

Considering the drawbacks of respiratory sampling techniques and the need for community-deployable mass testing with a minimal labor burden of trained personnel, the use of more flexible and less invasive specimens for COVID-19 diagnosis tests is crucial. Saliva, sputum, and throat washes are promising specimens for expanding and facilitating the testing sites due to the simplicity, safety, and non-invasive nature of their collection. During the early phase of SARS-CoV-2 infection, a relatively high viral load can be detected in samples collected from the oral cavity [3,4,5], indicating saliva as a promising alternative to NPS for routine COVID-19 screening. Saliva as a diagnostic specimen for detecting SARS-CoV-2 based on RT-qPCR has been tested with a positive agreement with NPS/RT-qPCR ranging from 75% to 92% [6,7]. Another bottleneck of the current SARS-CoV-2 RT-qPCR determination is RNA extraction. The elimination of the RNA purification step in molecular testing has demonstrated reliable detection of COVID-19 patients using both NPS and saliva samples [8,9,10].

An alternative to detection by RT-qPCR is reverse transcription loop-mediated isothermal amplification (RT-LAMP) [11,12]. RT-LAMP is a rapid, thirty-minute procedure that is more tolerant to inhibitors and can bypass a dedicated RNA extraction step. It is especially suited for PoC testing, as nucleic acid amplification is carried out at a constant temperature; therefore, this method does not require a thermal cycler and is compatible with a simple colorimetric readout [12,13,14,15]. The RT-LAMP of RNA purified from COVID-19 patients’ NPS samples has an average sensitivity of 86% [12]. The RT-LAMP of saliva samples without RNA extraction identified infection with up to 94% sensitivity [15].

To overcome the major drawbacks of the current reference RT-qPCR methods, we developed a diagnostic method based on nucleic acid amplification for detecting SARS-CoV-2 RNA in saliva without RNA extraction (Figure 1A). In addition, we compared different saliva-sampling techniques used for SARS-CoV-2 detection. In parallel to NPS, it was collected from the oral cavity with nylon or rayon swabs or as passive drool. The saliva specimens were stored in an RNA stabilization buffer and heat-treated to release RNA [16] and reduce the risk of infection during sample handling. The heat-treated and stabilized saliva specimens were analyzed directly, without the RNA extraction step, to simplify and shorten the RNA detection process. The samples were analyzed simultaneously with two techniques: RT-qPCR and RT-LAMP. In order to evaluate the robustness of the saliva sampling approach, the samples were tested in different laboratories, each running an in-house protocol. Saliva-based SARS-CoV-2 results were compared with NPS-paired saliva results from more than 700 individuals. The self-sampling method, such as collecting saliva, opens an opportunity to perform the sampling procedure in a decentralized manner and reduces the logistic burden and infection risk.

## 2. Results

### 2.1. Development of Saliva-Based SARS-CoV-2 Detection Method 

The diagnosis methods for SARS-CoV-2 rely on the detection of viral RNA by reverse transcription and amplification of the target segment of the viral genome from a specimen. A saliva specimen represents a suitable alternative to NPS, as a high viral burden in saliva was determined in the early stage of the infection [5]. Therefore, we set up a modified, straightforward diagnostic framework involving the following steps: (1) collecting saliva in the RNA stabilization buffer developed in this study; (2) heat treatment of stabilized saliva, and (3) nucleic acid analysis using one-step reverse transcription coupled with nucleic acid amplification (RT-qPCR or RT-LAMP) (scheme, Figure 1B). One-step isothermal amplification—RT-LAMP—was selected as an alternative to one-step RT-qPCR, as it is faster and can be adopted [17] as a PoC screening method. 

Three groups of subjects were studied: hospitalized COVID-19 patients (79), symptomatic and asymptomatic individuals from a public COVID-19 testing site (over 740), and individuals with no signs of infection and healthcare workers (over 750) (Figure 1A). We provided written step-by-step instructions, accompanied with an instructional video, to guide the individuals through the saliva self-collection procedure, with an emphasis on safety, appropriate time for sampling, amount of collected saliva, and after-sampling procedure. The NPS and saliva samples (as pairs) were collected at the COVID-19 testing site. For mass testing of individuals without symptoms, initially, only saliva specimens were collected. For individuals whose saliva specimens tested positive for SARS-CoV-2, the NPS was collected to confirm the infection. 

We compared three saliva collection techniques: oral cavity swab with nylon-tipped applicator, oral cavity swab with rayon-tipped applicator, and passive drool. At the collection site, to simplify the downstream processing, the saliva specimens were stabilized with an equal volume of RNA stabilization buffer and later heat-treated for 15 min [16]. The heat treatment of saliva assured biosafety during sample processing and analysis [18], the release of target RNA from the virions, and the inactivation of nucleases, which improved RNA preservation [19]. Furthermore, we compared different experimental setups to validate the robustness of the developed test (Appendix A).

### 2.2. Detection of Viral RNA at Low Concentrations

First, the analytical specificity and limit of detection (LoD) for RT-LAMP and RT-qPCR were determined. Based on the results of six benchmarked SARS-CoV-2 specific primer sets for RT-LAMP [12], the 3’ located on the N-gene encoding the nucleocapsid protein (NEB N2) and the envelope E gene (NEB E1) [12,20,21] were selected as the targets (Appendix A). Zhang et al. [20] designed the N2 and E1 RT-LAMP primer sets using an online software Primer Explorer V5. Human actin B (ACTB) or ribonuclease P (RNaP) mRNA was used as a control for the sample quality. For RT-qPCR, N1 and N2 targeting two regions of the N gene were selected as the targets (https://www.cdc.gov/coronavirus/2019-ncov/lab/rt-pcr-panel-primer-probes.html, accessed on 10 September 2020) [10,16,22] and human RNaP or 18SRNA (18S) RNA was used as a sample quality control (Appendix A).

The SARS-CoV-2 virus is rapidly evolving, generating new subtypes with nucleotide substitutions, which also appear in primer-or probe-binding regions. The temporal resolution assumes a nucleotide substitution rate of 8 × 10^−4^ substitutions/site/year (approximately 24,600 substitutions/genome/year, https://nextstrain.org/ncov/gisaid/global?l=clock, accessed on 23 October 2021) [23] with mutations concentrated within hotspots, one of which is the S1 protein. This can significantly alter the sensitivity and specificity of the PCR assay. Inclusivity of the COVID-19 RT-LAMP and RT-PCR tests was demonstrated by in silico analysis of the assay against all publicly available SARS-CoV-2 strains using the assay’s primers and probes. A total of 53,251 (from 21 March to 20 April 2021) and 47,807 (from 7 September to 7 October 2021) SARS-CoV-2 sequences [Global Initiative on Sharing All Influenza Data (GISAID, https://www.gisaid.org, accessed on 23 October 2021)] were aligned with the primers and probes used in this study (Appendix A). The in silico specificity analysis results indicated that the chosen primers could detect the currently circulating viral subtypes.

To monitor RT-qPCR amplification, we used TaqMan probes specific to the amplified targets. The isothermal amplification efficiency of the RT-LAMP reaction in real time was determined using Syto9 fluorescent dye, which had no impact on amplification within the given concentration range (Figure 2A, Appendix A). Furthermore, the fluorescent dye facilitated a melting curve analysis of the amplified product and, consequently, confirmation of the specificity of amplification and possible identification of false-positive products (Figure 2A, Appendix A). The RT-LAMP amplicons could be detected as an end-point colorimetric or fluorescence measurement or imaging (Appendix A) [24]. However, the end-point measurement fails to provide Tp and Tm values, which are essential for discriminating between true- and false-positive amplification. To determine the lower LoD and set the amplification conditions, we validated the performance of methods using synthetic RNA standard (Twist), whole virus material (EQA scheme INSTAND), and donor viral RNA diluted with heat-treated saliva, which was confirmed to be negative for SARS-CoV-2 (Figure 2B–F). The number of viral RNA copies was determined with digital PCR and used to estimate the copies per reaction. The RT-LAMP LoD for the N2 primer set with the WarmStart enzyme mix in 20 µL reaction volume was approximately 300 RNA molecules in saliva (Figure 2B), which is within the range reported by Kellner et al. [20]. The LoDs for N2 and E1 primer sets with the Isothermal Master Mix in 15 µL reaction volume were 119 and 70, respectively (Figure 2C).

A singleplex with the N1 primer–probe set and multiplex RT-qPCR with N1 and N2 primer–probe sets detected the viral RNA at approximately 10 RNA copies per reaction (Cq < 40) (Figure 2D,E). The analytical sensitivity of RT-qPCR with Ultraplex and AgPath-ID enzyme Master Mix was within 10 RNA copies, with no significant variation between N1 and N2 primer–probe sets (Figure 2E,F). The LoD values already provided the first evidence that the saliva-RT-LAMP is less sensitive than saliva-RT-qPCR for SARS-CoV-2, which was expected based on the literature [12].

In RT-qPCR with Ultraplex and AgPath-ID enzyme Master Mix, higher Cq values were obtained with the N2 primer–probe set compared to N1. This difference was taken into account when determining the Cq cut-off value for the S-5/15 (AgPath mix) protocol. For this protocol, Cq values obtained during the repeatability runs of method validation were carefully checked in the area close to the LoD, and the range of the highest Cq values was determined and rounded to the half value above. In order to account for the potential difference in the thresholds chosen between runs, 0.5 was added to this Cq value to obtain the cut-off value. This cut-off value was also confirmed on the real saliva samples. For the N1 region, the highest Cq value was 41.73, which was rounded to 42.0, and 0.5 was added, resulting in a cut-off value of 42.5. For the N2 region, extremely high Cq values were obtained in the repeatability study, which gave a cut-off value of 45.5. This was additionally checked in the analysis of real saliva samples, where such high values were observed in samples where N1 gave a negative result. In order to minimize the possible false-positive interpretation of the results, the Cq value of the last positive sample from the repeatability study was used for the determination of the cut-off value for the N2 assay (43.5).

In addition to analytical sensitivity, the occurrence of false positives was also determined to assess specificity. The saliva obtained from confirmed SARS-CoV-2 negative individuals was used to determine the analytical specificity for RT-qPCR and RT-LAMP. Out of the 160 reactions, nine were determined positive using an N2 primer set with RT-LAMP, yielding an analytical specificity of 94%. In contrast, for a singleplex with N1 primer–probe sets and N1, N2-multiplex RT-qPCR, analytical specificity was determined at 96% (7/189) and 98% (1/64), respectively. 

### 2.3. Stabilization of RNA in Saliva 

The pH variability of saliva presents a critical parameter in the end-point detection of RT-LAMP [15]; furthermore, saliva components and host RNases can impact the nucleic acid amplification efficacy. To circumvent the problem of saliva heterogeneity, to unify and increase saliva pH, and to further stabilize viral RNA, we developed a sample buffer for RNA stabilization, which contained chemical nuclease inhibitors tris(2-carboxyethyl)phosphine (TCEP) [25], EDTA, and TRIS buffer at pH 8.8. The addition of ion-exchanger Chelex [26] and RNase inhibitor RNAsecure into the sample buffer improved the amplification efficiency compared to that of water or even the sample buffer alone. The final mixture containing the sample buffer, Chelex, and RNAsecure was named the RNA stabilization buffer and used to stabilize the saliva specimens for RT-LAMP and RT-qPCR (Figure 3A). Furthermore, we examined whether Tween 20 [25], Proteinase K, guanidine chloride, and magnesium ions [26] improved RT-LAMP amplification. Guanidine chloride significantly extended the time to detect the amplification products and had a prolific impact on the melting curve (Figure 3B). However, Proteinase K, Tween 20, and magnesium ions did not affect the amplification; therefore, we did not include these chemicals in the RT-LAMP reaction. 

### 2.4. Impact of Saliva-Sampling Techniques on the Diagnostic Sensitivity of One-Step RT-LAMP

The diagnostic sensitivity of nucleic acid amplification was compared among three saliva-sampling techniques: oral cavity swab with nylon-tipped applicator, oral cavity swab with rayon-tipped applicator (Figure 4A), and passive drool (Figure 4B). At the COVID-19 testing site, along with an NPS, a sample of saliva was self-collected and stabilized with the RNA stabilization buffer. The saliva samples were then heat-treated and analyzed for SARS-CoV-2 without additional RNA extraction. 

The saliva sampling technique significantly affected the diagnostic sensitivity of RT-LAMP (LAMP-2/20 protocol) (Figure 4C–E, Appendix A). Among all positive samples, 10% (rayon swab) (Figure 4C) and 36% (nylon swab) (Figure 4D) samples tested positive for RT-LAMP. The best sensitivity was obtained for saliva samples collected as passive drool; the results for this saliva matched NPS positive results for RT-LAMP in 63% of the cases (Figure 4E). 

The samples were additionally analyzed using N2 and E1 primer sets and RT-LAMP Isothermal Master Mix (LAMP-5/15 protocol) (LoD values depicted in Figure 2C). The N2 and E1 primer sets’ RT-LAMP diagnostic sensitivities were 74% and 70%, respectively, and 83%, considering the readout being positive if at least one target was detected positive (Appendix A). The improved diagnostic sensitivity of the isothermal RT-LAMP test is attributable to the lower LoD values compared to WarmStart RT-LAMP. The diagnostic specificity was similar for all three saliva collection methods and was greater than 93% for RT-LAMP.

### 2.5. Validation of Saliva-Based Extraction-Free One-Step RT-qPCR for Diagnosis of SARS-CoV-2 Infection

Our SARS-CoV-2 diagnostic approach using a self-collected saliva specimen increases accessibility for testing and further eliminates the need for RNA extraction, saving time and resources. The NPS-paired saliva specimens were analyzed with RNA extraction-free one-step RT-qPCR. The results were then matched with the diagnostic NPS-RT-qPCR. Initially, a singleplex RT-qPCR (S-2/20) was used to analyze 85, 500, and 190 rayon oral cavity swabs, nylon oral cavity swabs, and passive drool specimens, respectively (Figure 5F,G, Appendix A). The passive drool specimens exhibited 95% diagnostic sensitivity and 100% specificity. Of the 142 confirmed SARS-CoV-2 positives, 123 (87%) nylon swabs were determined to be positive with one-step RT-qPCR. The rayon swabs exhibited the lowest sensitivity of 47%. 

The same saliva samples collected as passive drool were then reanalyzed by multiplex RT-qPCR using N1 and N2 primer–probe sets with either 5 μL sample in 20 µL reaction volume (M-5/20) or 2.5 µL sample in 10 µL reaction volume (M-2.5/10) (Figure 5A). The ability to evaluate several independent viral regions in a single reaction gave multiplex RT-qPCR an important advantage over a single target assay. A result of the multiplex was considered positive if one or more targets were detected and negative if no targets were detected. No clear correlation was observed between the Cq values of NPS and matched saliva samples (Figure 5A), confirming previous observations where viral loads in the NP cavity differed from those in the oral cavity [27]. The average Cq values of saliva-singleplex and multiplex RT-qPCR were higher compared to those of NPS-RT-qPCR (Figure 5B), suggesting lower viral load or less efficient RNA amplification. The RT-qPCR showed 95% and 98% sensitivity for multiplexes (M-5/20) and (M-2.5/10), respectively. There was no significant difference in sensitivity over a specific range of Cq values (Figure 5C–E, Appendix A). The direct comparison of singleplex (S-2/20) Cq values showed very good correlations with the matched Cq^av^ values of multiplexes (M-5/20) and (M-2.5/10) (Appendix A), as expected. To determine the impact of primer–probe selection, two multiplexes—(M-2.5/10) amplifying E1 and N2 or N1 and N2—were performed simultaneously for the selected saliva specimens (Appendix A). No difference in Cq values was observed, proving that different combinations of primer–probe sets yielded similar results. 

In order to evaluate the robustness of saliva one-step RT-qPCR, the passive drool samples were reanalyzed by singleplex RT-qPCR using the AgPath-ID Master Mix. A 5 µL of 5× diluted sample in 15 µL reaction volume (S-5/15) was used (Figure 5F). Furthermore, the analysis was performed at different locations. The average pooled Cq values for the N1 and N2 primers were higher than those for NPS amplification. The diagnostic sensitivity for (S-5/15) RT-qPCR was 96%, with diagnostic specificity above 99% (Appendix A). 

### 2.6. Sample Pooling

If the prevalence of viral infection among the tested individuals is low, then sample pooling might be a suitable strategy to minimize the testing costs [28]. Generally, NPS or saliva in the first days of infection yields approximately 10^5^ viral RNA molecules per milliliter [29,30]. In our study, the saliva-based one-step RT-qPCR LoD was approximately 10 RNA molecules per test (Figure 2), which translates into approximately 5000 molecules per milliliter of saliva. In sample pooling combining five samples, a five-fold dilution would, therefore, still contain enough RNA to yield a positive result. In order to assess the saliva sample pooling, the randomly selected known positives were combined with four negative samples to create five sample pools. The selected specimens had Cq values between 15 and 40, as determined from the saliva samples with the saliva-based one-step RT-qPCR S-2/20 protocol. On average, the Cq value in the pooled samples increased for 4.6 cycles. However, the pooled samples with a Cq > 35 were not detected, confirming a loss of assay sensitivity due to sample dilution within pooling (Figure 5G). This indicates that, with some limitations, a pooling strategy can be used to minimize the testing costs; however, the samples with Cq values above 35 would be missed. Nevertheless, the problem with sample dilution caused by sample pooling can be circumvented with an increase in the reaction volume. 

### 2.7. Saliva-Based Extraction-Free One-Step RT-qPCR for SARS-CoV-2 at PoC Locations

Finally, we tested the performance of saliva-based SARS-CoV-2 detection at PoC locations (working facilities). The saliva samples were self-collected from individuals with no signs of infection twice per week during a period of three months to identify asymptomatic virus carriers (Appendix A). We performed 769 RT-qPCR or RT-LAMP tests and determined 35 positive individuals for SARS-CoV-2, of which 33 were later confirmed to be SARS-CoV-2 positive by NPS-RT-qPCR. 

Next, we selected two households with documented close contact with a SARS-CoV-2-infected individual. The saliva specimens were self-collected at the location and stabilized with the RNA stabilization buffer according to the written instructions. For five individuals, we could detect viral RNA in the saliva before the appearance of clinical COVID-19 symptoms (Figure 6A,B), which confirmed that saliva-based RT-qPCR and RT-LAMP could detect asymptomatic individuals. The viral infection was corroborated by the NPS specimen analysis. The presence of the virus in saliva during the infection was monitored. The viral load peaked within the first week of infection detection and receded within 7 to 10 days. All individuals tested negative for viral RNA within 2 weeks after the first positive test. The RT-LAMP analysis was performed simultaneously with RT-qPCR, which confirmed the viral infection but failed to detect a lower viral load in saliva, which was weak after the onset of COVID-19 symptoms.

Furthermore, a saliva-based one-step RT-qPCR or RT-LAMP test was conducted to analyze the saliva of hospitalized COVID-19 patients. Most patients were hospitalized, on average, 10 days after COVID-19 symptoms first appeared. Among the 79 passive drool or nylon swab saliva samples, 52 tested positive with either the RT-qPCR or RT-LAMP test. No correlation was identified between the date of self-declared clinical COVID-19 symptoms or the date of admission to the hospital and saliva-based one-step RT-LAMP or RT-qPCR results (Figure 6C,D). This indicated that, during a later stage of infection [31,32], the saliva specimen is not reliable for detecting SARS-CoV-2 infection. Therefore, more reliable detection methods, including computed tomography scans of the chest or serological techniques, should be used.

## 3. Discussion

This study aimed to develop a rapid, straightforward, and reliable test for SARS-CoV-2 infection suitable for mass testing and requiring minimal sample manipulation. Multiple studies have demonstrated that the early detection of SARS-CoV-2 infection, accompanied by enhanced epidemiological surveillance strategies and isolation of SARS-CoV-2 positive contagious individuals, contributes to the prevention of future COVID-19 outbreaks. The mass testing for SARS-CoV-2 infection is, therefore, a critical step in limiting the viral spread. Although NPS is considered an optimal specimen type for detecting SARS-CoV-2, the saliva specimens yielded higher detection rates than NPS during early infection for most types of viruses [33,34], including SARS-CoV-2 [27,35,36,37,38]. Therefore, saliva is a highly desirable specimen, which can also be easily collected. Young et al. [39] demonstrated that smartphone screen swab samples could also be used for detecting positive individuals.

We assessed three saliva self-sampling techniques (rayon and nylon oral cavity swabs and passive drool) and compared them using two viral RNA amplification methods (RT-LAMP and RT-qPCR). We showed that the saliva sampling technique is an important step in determining the diagnostic sensitivity of SARS-CoV-2 RNA amplification. The highest RNA amplification sensitivity was achieved with saliva collected as a passive drool. Oral cavity swabs using rayon-and nylon-tipped applicators were less effective than passive drool, which is probably due to the less efficient sample mixing with the RNA stabilization buffer and the lower volume of the collected sample. Additionally, the critical points of saliva self-sampling were (i) the amount of collected saliva and (ii) sampling time. The amount of saliva should not exceed three times that of the RNA stabilization buffer to be effectively preserved for a longer period after heat treatment. Furthermore, the saliva collected immediately after the consumption of food is not a reliable specimen. 

Several studies have demonstrated that some types of RNA stabilization buffer that simultaneously inhibit RNA degradation and promote RNA release can significantly improve the diagnostic sensitivity of RT-LAMP or RT-qPCR [10,15,25,26]. We developed an RNA stabilization buffer that is compatible with nucleic acid amplification tests, meaning that the saliva specimens can be analyzed without RNA extraction before the RT-LAMP and RT-qPCR tests [16]. The RNA stabilization buffer was used to preserve RNA in saliva against RNases after virion lysis [15]. The buffer, which is composed of TCEP, chelator EDTA, ion-exchanger Chelex, and heat-activated RNase inhibitor RNAsecure, downgrades the viscosity of saliva and inhibits RNA degradation. The ion-exchanger Chelex, in combination with DTT, was also used by Howson et al. [26], who demonstrated that Chelex combined with heat treatment could significantly improve the detection of viral RNA with RT-LAMP. Furthermore, this buffer is compatible with colorimetric LAMP detection, as it increases the pH value of the sample to above eight.

The saliva mixed with the RNA stabilization buffer was further heat-treated to inactivate the virus, which minimizes the risk of infection during sample processing and promotes the release of RNA from virions and cells. The compatibility of the buffer with the amplification methods enabled the analysis of the saliva without RNA extraction, which significantly shortened the detection time and simplified sample processing. 

Among the nucleic acid amplification tests, extraction-free one-step RT-qPCR is superior in many aspects [40], such as its high dynamic range and high specificity. Although RT-LAMP is not as sensitive as RT-qPCR, RT-LAMP is especially suitable for PoC testing, as nucleic acid amplification is carried out at a constant temperature, eliminating the need for a thermal cycler [13,41]. The LoD determined here or that obtained from the literature [12,40] indicates that RT-qPCR is more sensitive than RT-LAMP. In order to determine the diagnostic sensitivity and specificity of saliva-based extraction-free one-step RT-qPCR and RT-LAMP, more than 700 saliva samples were collected simultaneously with NPS specimens at an official COVID-19 testing site and analyzed by RT-qPCR and RT-LAMP. The diagnostic sensitivity was between 67% (N2 gene) and 83% (N2 and E1 genes) for the RT-LAMP of the passive drool specimens. The sensitivity and specificity of the passive drool saliva-based RT-qPCR test without RNA extraction were 95% and 100%, respectively. 

Since a great advantage of one-step RT-qPCR testing without extraction of viral RNA is the speed of analysis, the results, where only one viral target was amplified, were interpreted as positive, so the self-isolation of the potentially positive individual can start without delay. The individual is then invited to resubmit a new saliva sample for a follow-up analysis.

For RT-qPCR, N1 and N2 primer–probe sets amplifying different regions of the N gene were selected according to the CDC guidelines and based on Vogel et al.’s report [22], demonstrating very high sensitivity for these primer–probe sets. However, comparable results were obtained using a combination of E1 and N2 primer–probe sets. The primer and probe sequences were analyzed in silico against SARS-CoV-2 strains in the GISAID database from 21 March to 20 April 2021 (low prevalence of Delta strain) and from 7 September to 7 October 2021 (high prevalence of Delta strain). The higher mismatch frequencies for individual targets were identified for the N1_F primer (20%) and RdRP_F (100%) (high prevalence of Delta strain). Due to the high rate of mutations (in the range of 10^−3^ per site per year [42,43]), it is essential to monitor the mutation frequencies for individual targets closely.

## 4. Conclusions

Our study shows that the saliva collected as passive drool can be considered a replacement for NPS for mass testing at COVID-19 testing sites, except for hospitalized COVID-19 patients, due to the low viral load in their saliva in the later stages of the infection. Furthermore, the saliva stored in the RNA stabilization buffer and heat-treated can be used in nucleic acid amplification tests without the RNA extraction step, which considerably reduces the labor costs and shortens the time from sample collection to diagnosis. For mass testing of populations with low SARS-CoV-2 infection prevalence, sample pooling can be introduced in diagnostics to reduce resource consumption [28]. Although RT-LAMP is a less favorable method than RT-qPCR due to its reduced sensitivity, it is a suitable PoC method in which qPCR machines are not available. 

The saliva screening approach has several advantages: the use of saliva specimens eliminates uncomfortable NPS sampling, and the self-collection of saliva can diminish the exposure of healthcare personnel to viral infection and is a non-invasive procedure that requires fewer consumables [44]. Furthermore, self-collection of saliva as passive drool is easy to perform, does not require a particular setup, and takes around one minute. The self-collected saliva can be delivered to the collection point, which is a convenient approach for establishing mass testing. The saliva-based extraction-free one-step RT-qPCR does not require an RNA extraction procedure; therefore, the net cost for the test is lower than that for the NPS-RT-qPCR test. Additionally, staff costs are lower, as, in most cases, sampling can be performed by the individual. Since RNA extraction is not required, the one-step RT-qPCR reaction can be completed within one hour. The pre-mixes of the RT-qPCR reaction can be prepared in advance and, later, samples can be added to pre-pipetted reaction mixtures. Automation can further speed up the process. 

## 5. Materials and Methods

### 5.1. Ethics

The study was reviewed and approved by the National Medical Ethics Committee of Slovenia (0120-5/2020/2), and all methods were carried out following relevant guidelines and regulations. The following is the list of study participants: (i) symptomatic and asymptomatic individuals, (ii) COVID-19 patients, and (iii) individuals without symptoms of infection. They were informed in writing about the purpose and procedure of the study and consented to provide the saliva sample.

### 5.2. Specimens

NPS-paired and self-collected saliva samples were collected at a public testing station organized by the Emergency Medical Service, Jesenice, Slovenia (testing site). The NPS was collected for routine SARS-CoV-2 testing by trained staff. The NPS samples were transported in a 3 mL commercial viral transport medium (HiViral Transport Kit, Ref. MS4760-100NO, HIMEDIA) and subjected to routine clinical testing on the same day in the National Laboratory of Health, Environment, and Food (NLHEF). 

The saliva was self-collected either as a passive drool (~100 µL) into 1.5 mL microcentrifuge tubes with 100 µL of the RNA stabilization buffer, or by using swabs—FLOQ Swabs, regular molded nylon swab (520CS01, Copan Italy, Brescia, Italy), or rayon-tipped swab (155C, Copan Italy, Brescia, Italy).

For self-collection of the saliva specimens, we provided detailed guidelines accompanied by a video. We instructed the individuals that the saliva samples should be collected in the morning or at least 30 min after consumption of food or drink, the amount of saliva should be between 100 and 200 µL (indicated as 3–5 drops of saliva), sampling should be performed in isolation to minimize the spread of infection, and after sampling, the sample tube should be disinfected and refrigerated.

The passive drool or oral cavity swab saliva samples obtained from COVID-19 patients were additionally collected as part of the routine procedures at the Clinics for Infectious Disease and fever Conditions, University Medical Center, Ljubljana, and University Clinic of Pulmonary and Allergic Diseases Golnik. The samples from these two clinics were not paired with the NPS. Information on the beginning of symptoms was collected as patient self-declaratory data.

Passive-drool saliva (~100 μL) was self-collected from individuals with no symptoms of infection.

### 5.3. Saliva Sample Treatment

Passive-drool saliva (~100 µL) or oral cavity swabs was stabilized with RNA stabilization media (1:1). RNA stabilization media contains 30 µL of 10× sample buffer (25 mM TCEP (Tris(2-carboxyethyl)phosphine), 5 mM EDTA (pH 8), 100 mM TRIS, pH 8.8), 70 µL Chelex100 (10% (*w/v*), BioRad), and 4 µL RNAsecure (Invitrogen). At the collection site, the saliva collection tubes were decontaminated with 70% ethanol, delivered to the diagnostic laboratory within a few hours, and stored at −20 °C until further processing. All saliva specimens were subjected to 5 to 15 min heat inactivation at 95 °C [18,45], cooled on ice, and immediately analyzed or stored at −80 °C. Until heat inactivation of the virus, the samples were processed in a biosafety level-2 cabinet.

### 5.4. RT-qPCR Saliva Testing

Four saliva-based extraction-free one-step RT-qPCR tests for SARS-CoV-2 were conducted (summarized in Appendix A). The primer pairs and probes used for single and multiplex RT-qPCR targeting two regions within the N gene (N1, N2), human RNaP, or 18SrDNA, together with their final concentrations, are listed in Appendix A [22].

For S-2/20, M-5/20, M-2.5/10 protocols, we used the enzyme mixture Ultraplex 1-Step 4×ToughMix (QuantaBio, Beverly, MA, US). The simultaneous detection of multiple targets, namely SARS-CoV-2 N1 and N2 and hRNaP or single gene (SARS-CoV-2 N1 amplicon), was performed on a Light Cycler 480 Real-Time PCR System (LC480, Roche, Basel, Switzerland). DNA oligonucleotides were resuspended at 100 µM concentration. RT-qPCR (S-2/20) as the singleplex reaction was carried out in a 20 μL reaction volume with 2 μL of a sample. Multiplex RT-qPCR (M-2.5/10) and (M-5/20) reactions were performed in 10 or 20 μL reaction volumes with up to 2.5 or 5 μL heat-inactivated stabilized saliva, respectively. The LC480 software was used for the data analysis and detecting the fluorescence intensity. The sample was considered positive if at least one target RNA was detected with Cq values of <40 for N1 and N2. The Cq values for the RNaP assay were used for controlling the sample presence, and samples with no amplification of RNaP were excluded from further analysis.

An AgPath-ID One-Step RT-qPCR Mix (Ambion, Austin, TX, USA) was used for the singleplex (S-5/15) RT-qPCR protocol. A 5 µL of 5× diluted sample in 15 µL reaction volume was used. All samples were analyzed with N1 and N2 amplicons, and for internal control, 18S rDNA was used. A result was considered positive if any of the viral targets were amplified. RT-qPCR was carried out in 384-well plates (Applied Biosystems, Waltham, MA, USA), with the reactions run in duplicate on an ABI 7900 HT Fast (Applied Biosystems, Waltham, MA, USA). The SDS 2.4 software (Applied Biosystems, Waltham, MA, USA) was used for fluorescence acquisition and data analysis. The samples were considered positive if a target RNA was detected repeatedly (in at least two reactions out of four), with Cq values of < 42.5 for N1 and < 43.5 for N2. These Cq cut-off values were set based on the results of the repeatability runs of method validation (Figure 2F) and from analysis of the real saliva samples. The samples with no amplification of 18S rDNA were excluded from further analysis. An NTC and positive control were included in each RT-qPCR run in all laboratories to monitor the possible contamination of the PCR reagents.

### 5.5. RT-qPCR Assay for NPSs

A confirmed case of COVID-19 was identified as an individual with NPS who tested positive for SARS-CoV-2 using laboratory-based RT-qPCR. The NPS specimens were analyzed at the NLHEF. Swabs in VTM were vortexed, and 140 μL of VTM solution was used for viral RNA extraction using the automated nucleic acid extraction instrument QIAcube Connect (QIAGEN, Germantown, MD, USA) and QIAamp Viral RNA Mini Kit (QIAGEN, Germantown, MD, USA) following the manufacturer’s instructions. The procedure resulted in the extraction of 50 μL RNA. SARS-CoV-2 RNA was detected by targeting the E and RdRP genes, as described by Corman et al. (2020) [40]. For both assays, 20 μL reactions, including 5 μL of extracted viral RNA, 4 μL of 5× master mix and 0.1 μL of RT enzyme (both Roche LightCycler Multiplex RNA Virus Master, Basel, Switzerland), 0.5 μL LightMix Modular SARS-CoV primer–probe mix for a respective gene (E-gene and RdRP, both TIB MOLBIOL), and 10.4 μL of PCR-grade water, were used for the RT-qPCR reaction. PCR was run on an RGQ PCR cycler (QIAGEN, Germantown, MD, USA). The primer and probe sequences are shown in Appendix A, and the cycling conditions are listed in Appendix A. Samples with Cq < 38 were considered positive. Detection of human RNaP mRNA was used as a control for sample quality. In each viral RNA extraction set and in each PCR reaction, negative controls were used and treated in the same way as a sample.

### 5.6. Asymptomatic and Symptomatic Criteria

A symptomatic patient was identified as one with symptoms such as fever, cough, and sore throat. An asymptomatic individual was identified as having positive SARS-CoV-2 test results without any relevant clinical symptom. 

### 5.7. RT-LAMP Saliva Testing

An RT-LAMP test named LAMP-2/20 was assembled in 20 µL reaction volume. LAMP Master mix was prepared on ice immediately before use. Per reaction, it contained 10 µL of the WarmStart Colorimetric RT-LAMP 2× Master Mix (M1800, New England Biolabs, Ipswich, MA, USA), 2 µL of the 10× LAMP primer mix (Appendix A) [46,47], 0.4 µL Syto9 (50 µM, Invitrogen, Waltham, MA, USA), and 1 µL of 2× sample buffer and filled with nuclease-free water (4.6 µL). Primers, depicted in Appendix A, were used in final concentrations of 0.2 µM for F3/B3, 0.4 µM for LF/BF, and 1.6 µM FIP and BIP. The LAMP Master mix (18 µL) was distributed into the wells before pipetting 2 µL of heat-treated saliva sample into each well on the plate. 

For the RT-LAMP test named LAMP-5/15, E1 and N2 primer sets amplifying SARS-CoV-2 targets and human ACTB primer set for sample control were used. The RT-LAMP reaction was performed in 15 µL reaction volume, which contained 7.5 µL of 2× Isothermal Master Mix (Opti-Gene, Horsham, UK), 0.6 µL of Transcriptor 5× RT buffer (Roche, Basel, Switzerland), and 1.5 U Transcriptor RT (Roche, Basel, Switzerland). The saliva sample was 5× diluted in the extraction buffer, and 5 µL was added to the reaction mix.

All solutions were prepared in nuclease-free water. An NTC was included in each RT-LAMP run to monitor the possible contamination of the reagents. A 96-well plate format was used, sealed with a thermostable optically clear adhesive seal, and kept on an ice-cold cooling block. The reactions were incubated in an LC480 at 63 °C for 35 min (LAMP-2/20) or 65 °C for 60 min (LAMP-5/15) with a lid heated to 110 °C. The fluorescence of each well was measured every 30 s. The reaction was stopped by heating it to 95 °C for 5 min, and then a melting curve analysis was performed from 60 °C to 98 °C. The LC480 SW 1.4.1. software (Roche, Basel, Switzerland) was used for the fluorescence acquisition and determination of the time of positivity (Tp) and melting temperature of the amplified product (Tm). The data were analyzed using the fit-point algorithm, and Tm was determined with Tm melting analysis.

The samples amplified by the LAMP-2/20 method were considered positive if a target RNA was detected with Tp values of <25 min for N2 amplicon (one replicate). In contrast, the samples amplified with the LAMP-5/15 method were considered positive if a target RNA was detected in at least two out of four reactions with Tp values of <25 min for both (E1 and N2) amplicons. For the Tm values, a positive result was only obtained if the measured Tm value corresponded to that of the positive control (±0.5 °C). The Tp and Tm values for the ACTB amplicon were monitored for the control of sample preparation. 

### 5.8. Mismatches in Primer- and Probe-Binding Regions

An in silico analytical specificity was performed by aligning all primer sequences against SARS-CoV-2 GISAID. We calculated the mismatch frequencies using the covidcg.org database (Appendix A). 

### 5.9. Positive and Negative Controls for Evaluating RT-qPCR and RT-LAMP Protocols

The dilution series of various positive controls were used for evaluating the RT-qPCR and RT-LAMP protocols. Synthetic SARS-CoV-2 RNA control (Twist Bioscience #102019) was diluted from 10,000 to 1 copy/µL in heat-inactivated stabilized saliva collected from healthy individuals. The dilution series of the whole virus material from the EQA scheme (INSTAND) were prepared in heat-inactivated stabilized negative saliva 5× diluted in the extraction buffer. The dilutions were freshly prepared before the analysis. Values were assigned to positive controls using a one-step RT-ddPCR advanced kit for probes (BioRad) with N1 and N2 primer–probe sets according to the manufacturer’s instructions. As negative controls, either nuclease-free water, extraction buffer, or heat-inactivated stabilized saliva was used.

### 5.10. Software and Statistics

The graphics and data were prepared and analyzed using Origin 2018 or GraphPad Prism 6 software. The NPS result was used as the reference standard to calculate the positive percent agreement (PPA) and negative percent agreement (NPA) for the NPS-paired saliva specimens. The Cq value was determined for each positive SARS-CoV-2 specimen by using the qPCR apparatus software. The parametric paired, two-tailed Student *t*-test was used to test the statistical differences between the paired samples. 

The diagnostic concordance rate is defined as the sum of the matched positive and negative saliva samples to NPS divided by the total number of samples. Sensitivity is the proportion of true-positive tests out of all COVID-19 positive individuals. Sensitivity is the percentage of true negatives out of all subjects tested negative for COVID-19.

### 5.11. Sample Pooling

Sample pooling of five samples (*N* = 5) was performed by combining equal volumes of saliva. 

## Figures and Tables

**Figure 1 molecules-26-06617-f001:**
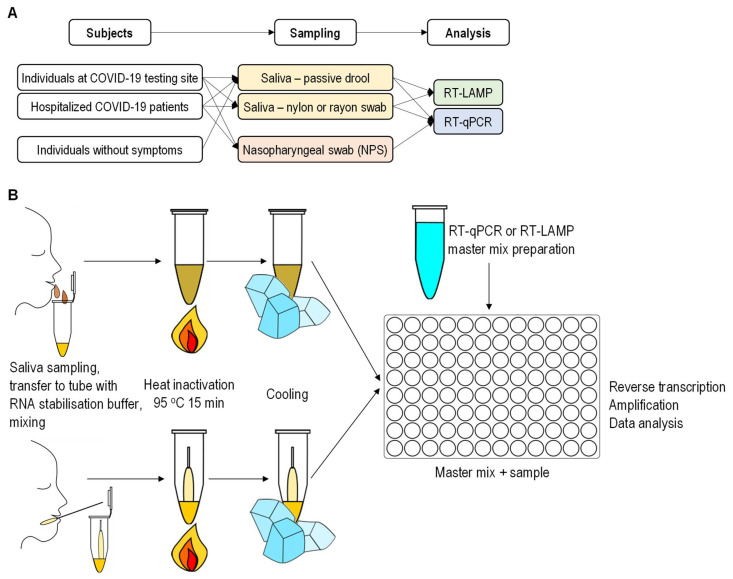
Diagnostic workflow for SARS-CoV-2 testing in saliva specimens. (**A**) Design of the study. Samples were collected from three groups of individuals: (i) individuals with the risk of being infected, (ii) COVID-19 patients admitted to a hospital and (iii) individuals without symptoms. Three saliva-sampling techniques were compared: passive drool and oral cavity swabs with nylon- and rayon-tipped applicators. NPS-paired saliva specimens were collected from asymptomatic and symptomatic individuals at the COVID-19 testing site. The specimens were analyzed for SARS-CoV-2 with RNA extraction-free one-step RT-LAMP or RT-qPCR. The NPS specimens were analyzed at the official testing laboratory National Laboratory of Health, Environment, and Food with a diagnostic RT-qPCR test, including RNA extraction. (**B**) Workflow of SARS-CoV-2 testing. Saliva as (i) an oral cavity swab or (ii) passive drool was collected in a tube with an RNA stabilization buffer. The specimen was heat-treated. After cooling, a one-step RT-LAMP or RT-qPCR test was conducted to detect SARS-CoV-2 in the saliva.

**Figure 2 molecules-26-06617-f002:**
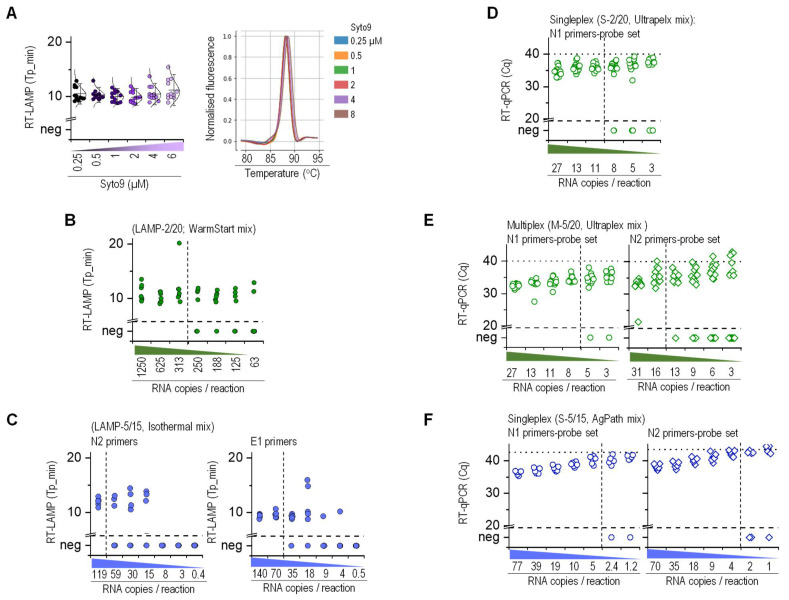
Analytical performance of saliva-based one-step RT-LAMP and RT-qPCR. (**A**) Effect of Syto9 fluorescent dye on RT-LAMP amplification using the N2 primer set and WarmStart Master Mix. The RNA diluted in water (315 copies per reaction) was amplified in 20 µL reaction volume (*n* = 11). The melting curves of the amplified target in the presence of Syto9 fluorescent dye are depicted. (**B**,**C**) LoD of RT-LAMP. (**B**) RNA (Twist) was amplified with N2 primer set in 20 µL reaction volume using the WarmStar Master Mix. (**C**) RNA (INSTAND) was amplified with N2 or E1 primer in 15 µL reaction volume using an Isothermal Master Mix. Serial dilutions of standard viral RNA were prepared in heat-treated SARS-CoV-2-negative saliva (*n* = 7 (**B**); *n* = 6 (**C**)). (**D**–**F**) LoD of RT-qPCR. (**D**,**E**) RNA (Twist) was amplified using Ultraplex Master Mix in 20 µL reaction volume as a singleplex with N1 primer–probe set (*n* = 20) (**D**) or as a multiplex with N1 and N2 primer–probe set (*n* = 16) (**E**). (**F**) RNA (INSTAND) was amplified using AgPath-ID Master Mix in 15 µL reaction volume as a singleplex with N1 and N2 primer–probe set (*n* = 16). (**D**–**F**) Serial dilutions of standard SARS-CoV-2 viral RNA were prepared in heat-treated SARS-CoV-2-negative saliva. A vertical dashed line indicates a lower LoD. A horizontal dotted line indicates the Cq cut-off value.

**Figure 3 molecules-26-06617-f003:**
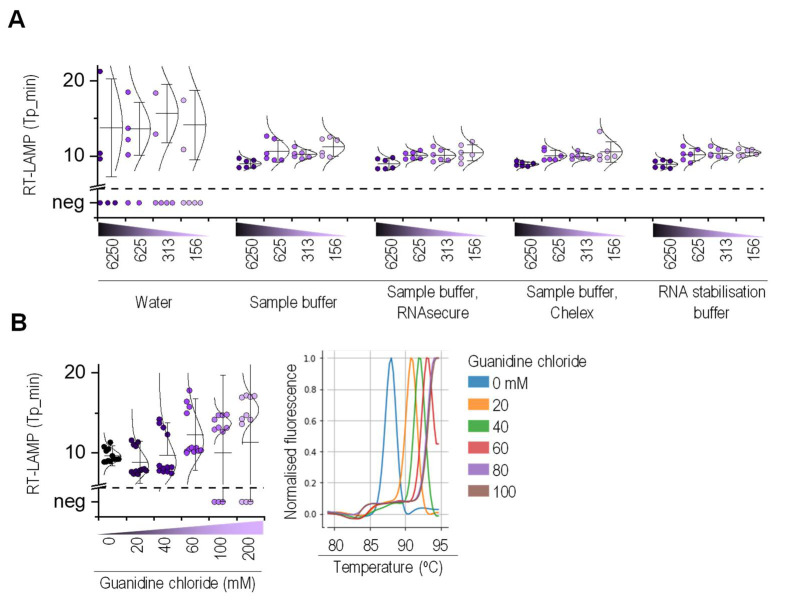
RNA stabilization buffer improved one-step RT-LAMP amplification. (**A**) A viral RNA diluted in heat-inactivated saliva from healthy individuals was mixed in 1:1 ratio with either water, sample buffer, sample buffer with RNAsecure or Chelex, or RNA stabilization buffer; heat-treated; and RT-LAMP-amplified (*n* = 6). (**B**) The RT-LAMP reaction mix with guanidine chloride was used to amplify RNA (315 copies per reaction) (*n* = 12). (**A**,**B**) RNA was amplified with N2 primers in 20 µL reaction volume using a WarmStart Mix.

**Figure 4 molecules-26-06617-f004:**
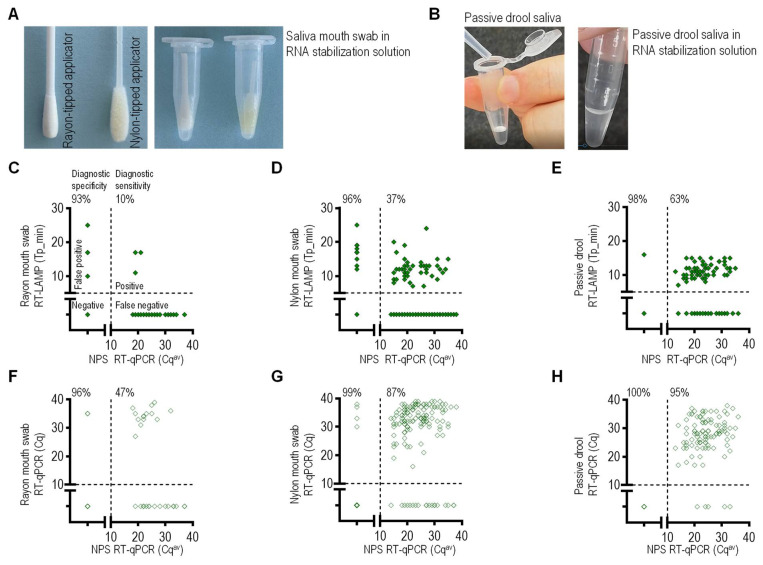
Passive drool outperformed oral cavity swabs in SARS-CoV-2 detection. (**A**) Oral cavity swab applicators with rayon or nylon tips. The swab tip was saturated with saliva and placed in a microcentrifuge tube with RNA stabilization buffer (100 µL). (**B**) Three drops of saliva (~100 µL) collected as passive drool were mixed with 100 μL of RNA stabilization buffer. (**C**–**E**) RT-LAMP and (**F**–**H**) RT-qPCR performances on saliva samples collected in parallel with NPS specimens. Saliva samples were self-collected in the RNA stabilization buffer. (**C**–**E**) Tp values of RT-LAMP are plotted against the corresponding Cq^av^ values of RT-qPCR for NPS. Cq^av^ was calculated as an average of the E1 and RdRT Cq values. A rayon (**C**) or nylon swab (**D**) or passive drool (**E**) saliva sample (2 µL) was added to the reaction mixture for a total volume of 20 μL, and RT-LAMP amplified using WarmStart Mix and N2 primer set. (**F**–**H**) Saliva-based RT-qPCR Cq values are plotted against the corresponding Cq^av^ of RT-qPCR for NPS. A rayon (**F**) or nylon swab (**G**) or passive drool (**H**) saliva sample (2 µL) was added to the reaction mixture for a total volume of 20 μL, and RT-qPCR amplified using Ultraplex Mix and N1 primer–probe set. Numbers above each plot indicate diagnostic specificity and sensitivity (for details, see also Appendix A).

**Figure 5 molecules-26-06617-f005:**
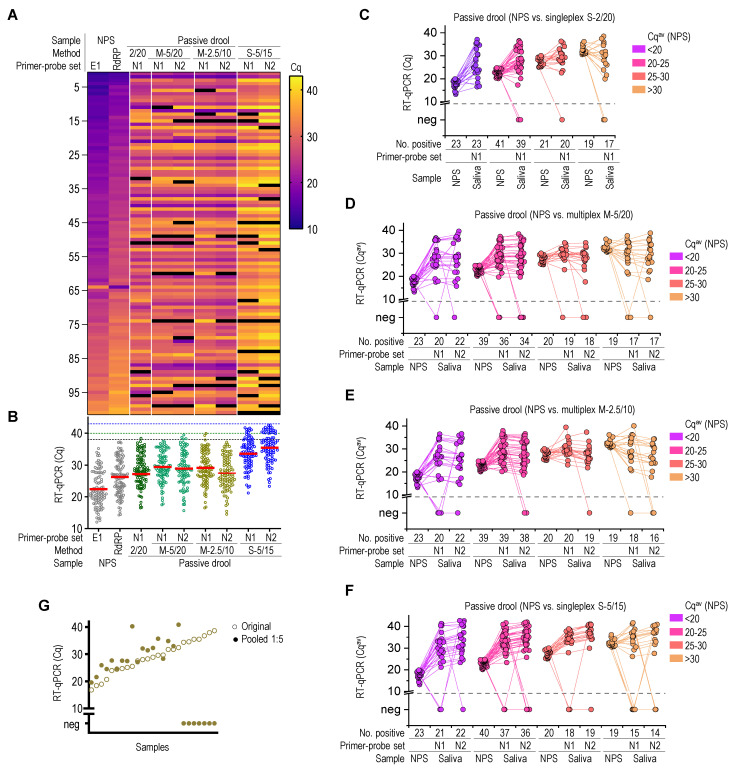
Diagnostic performance of extraction-free one-step RT-qPCR for SARS-CoV-2 in saliva. (**A**) Heat map of matched Cq values (black indicates Cq values above 43 and negative results). (**B**) Pooled Cq values with depicted average value as a red line. (**C**–**F**) The Cq value of single RT-qPCR and (**C**) Cq values of multiplex RT-qPCR (**D**–**F**) for passive drool and the corresponding Cq^av^ RT-qPCR of NPS were compared. (**G**) RT-qPCR analysis of pooled saliva samples. One microliter of selected saliva samples with Cq ranging from 15 to 40 was combined with five saliva samples negative for SARS-CoV-2 (each 1 µL) and amplified using the S-2/20 protocol. For each sample, the matched original and pooled Cq values were plotted.

**Figure 6 molecules-26-06617-f006:**
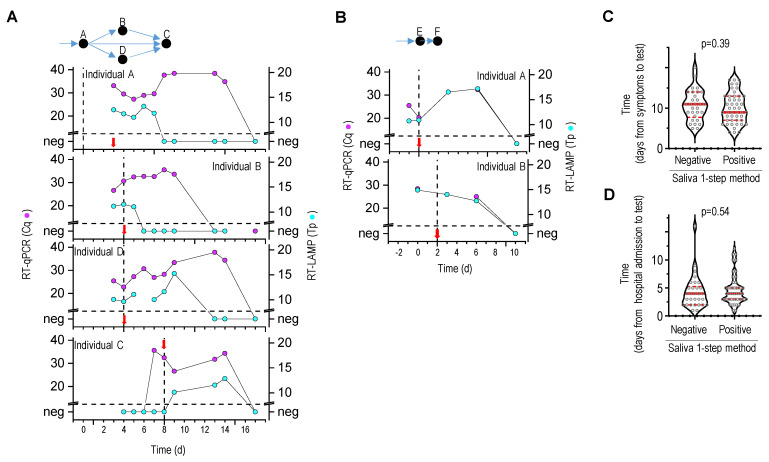
Longitudinal analysis of SARS-CoV-2 virus in infected individuals and hospitalized patients. (**A**,**B**) Sequential saliva sampling and detection of SARS-CoV-2 infection. Two households are shown. The appearance of COVID-19 symptoms is marked with a vertical dashed line, and a red arrow indicates the day of a positive NPS-RT-qPCR test. The viral infection among individuals is schematically presented at the top. (**C**,**D**) Saliva collected from hospitalized COVID-19 patients was analyzed for the SARS-CoV-2 virus. Results grouped as positive and negative for SARS-CoV-2 in saliva are presented in relation to the time between symptoms or hospital admission and saliva tests. (**A**–**D**) Saliva samples were analyzed with saliva-based one-step RT-LAMP or RT-qPCR, and combined results were presented.

## Data Availability

The data presented in this study are available on request from the corresponding author.

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
