# Peer review of "Robust Saliva-Based RNA Extraction-Free One-Step Nucleic Acid Amplification Test for Mass SARS-CoV-2 Monitoring"

_molecules, 2021, doi:10.3390/molecules26216617_

Round 1
Reviewer 1 Report
Authors of the provided article proposed a new method for COVID-19 testing, which allows elimination of the RNA isolation step, thereby reducing the analysis time. Saliva samples with RNA stabilized by a specially designed buffer were used for this purpose. The use of saliva instead of nasopharyngeal swabs as a material for testing allows to perform sampling procedures without trained personnel, reduces risk of infection and helps to avoid NPS contraindications. In the course of this work, methods for detecting COVID-19 based on RT-LAMP and RT-qPCR were developed.
The sensitivity and specificity of RT-qPCR protocols from 95% to 100% allow the proposed methods to be used as a simplified replacement for the standard NPS testing protocol for mass screening for COVID-19. The protocol is also convenient for detecting infection in the early stages before the appearance of clinical symptoms, for example, in persons who have been in contact with patients.
The disadvantage of the method based on RT-qPCR is that it requires a large number of PCR cycles (40 or more) for correct detection, since positive samples are considered those where target RNA was detected with Cq values <40 for protocols S-2/20, M- 5/20, M-2.5 / 10 and Cq <42.5 and 43.5 for S-5/15 protocol. The standard protocol for detecting viruses from NPS requires a Cq <38.
It is also not very clear why protocol S-5/15 used four replicates to define a sample as positive or negative, while for other protocols only one is mentioned. Similarly, for the RT-LAMP protocols. In LAMP-5/15 four replicates are used, positive results in two of four allow considering the sample as positive; for LAMP-2/20, replicates are not mentioned.
Author Response
Authors of the provided article proposed a new method for COVID-19 testing, which allows elimination of the RNA isolation step, thereby reducing the analysis time. Saliva samples with RNA stabilized by a specially designed buffer were used for this purpose. The use of saliva instead of nasopharyngeal swabs as a material for testing allows to perform sampling procedures without trained personnel, reduces risk of infection and helps to avoid NPS contraindications. In the course of this work, methods for detecting COVID-19 based on RT-LAMP and RT-qPCR were developed.
The sensitivity and specificity of RT-qPCR protocols from 95% to 100% allow the proposed methods to be used as a simplified replacement for the standard NPS testing protocol for mass screening for COVID-19. The protocol is also convenient for detecting infection in the early stages before the appearance of clinical symptoms, for example, in persons who have been in contact with patients.
The disadvantage of the method based on RT-qPCR is that it requires a large number of PCR cycles (40 or more) for correct detection, since positive samples are considered those where target RNA was detected with Cq values <40 for protocols S-2/20, M- 5/20, M-2.5 / 10 and Cq <42.5 and 43.5 for S-5/15 protocol. The standard protocol for detecting viruses from NPS requires a Cq <38.
The S-5/15 protocol was developed and evaluated by NIB, which is an expert in the field of plant pathology, GMO detection, and metrology (designated as EURL, NRL, holder of national etalon). Based on extensive experience in the introduction of methods for the low-level presence of targets, NIB has set the Cq values above 40 as positive. Setting the threshold should not be unique for all protocols as the quantification results can vary by thousands of fold when measuring viral genomes (https://doi.org/10.1016/j.ymeth.2021.03.016). Recently, a publication showing results from SARS-CoV-2 testing with qPCR has stated that the “Cq values should not be used for cut-off setting to stratify risk or guide analytical performance” (Dangers of using Cq to quantify nucleic acid in biological samples; a lesson from COVID-19 | Clinical Chemistry | Oxford Academic (oup.com)). Note that the primary difference between their protocol and those presented in this study lies in the analysis of crude samples, which means that we do not test purified RNA (as in the case in NPS samples). The higher difference in Cq values was expected and confirmed in a comparative study, when the extracted RNA was tested in parallel with the crude extract of saliva (spiked with whole virus material from INSTAND EQA (INSTAND material) and real infected sample). An average difference of approximately 3 Cq is mainly attributable to the later onset of amplification or less efficient amplification in a crude saliva sample, compared to the amplification of the extracted RNA. A similar effect was observed with dPCR, even though it is less sensitive to inhibitory substances, where we determined a three-times lower number of copies in the crude saliva in comparison to the extracted RNA. The principle for determining the last Cq value of the positive results was thus based on the results obtained from the evaluation of the protocols on spiked INSTAND material in saliva and analysis of real samples.
The Cq values obtained during the repeatability runs of method validation were carefully checked. In the area close to the LoD, where some wells started yielding »undetermined« (negative) results, the range of the highest Cq values was observed (not considering out-layers). The highest Cq value was rounded to the half value above, and then 0.5 was added to this Cq value to account for the potential difference in the thresholds chosen between runs. This procedure is part of the protocol for method validation in the GMO detection laboratory accredited according to ISO/IEC 17025. For the N1 region of the N gene, we obtained the highest Cq value as 41.73, which was rounded to 42.0 and 0.5 was added, resulting in a cut-off value of 42.5. In the case of N2 region, we obtained extremely high Cq values in the repeatability study, which gave a cut-off value of 45.5 (when using the same logic as that used for N1). However, when analyzing the Cq values from the real saliva samples, we concluded that such high values normally occur in only one replicate, and that in most of those samples, the N1 assay gave negative result. To minimize the possible false-positive interpretation of the results, we decided to use the rounded average of the Cq value of the last positive sample from the repeatability study as the cut-off value for N2 assay (43.5).
Just for illustration, a proportion of individuals with Cq > 38 was approximately 1 of 99 (S-2/20), 4 of 96 (N2 target M-5/20), 2 of 99 (N1 target M-2.5/10), 17 of 98 (N1 target S-5/15), and 26 of 98 (N2 target S-5/15).
Cq values vs. no. of RNA molecules per reaction for RT-qPCR S-5/15 protocol.
It is also not very clear why protocol S-5/15 used four replicates to define a sample as positive or negative, while for other protocols only one is mentioned. Similarly, for the RT-LAMP protocols. In LAMP-5/15 four replicates are used, positive results in two of four allow considering the sample as positive; for LAMP-2/20, replicates are not mentioned.
The NIC has been using only one replicate per RT-qPCR or RT-LAMP reaction (for protocols S-2/20, M-2.5/10, and M-5/20 LAMP-2/20). The rationale behind this decision was to identify the most cost-effective protocols for mass testing with high enough sensitivity (over 95%). The RT-qPCR S-2/20 protocol has already provided 96% diagnostic sensitivity for passive drool saliva samples. Later, we replaced S-2/20 for multiplex (two targets per viral genome, one replicate); the diagnostic sensitivity was similar to that obtained for singleplex. Considering cost-effectiveness, this proves that one technical replicate can also provide sufficiently good results. In contrast, RT-LAMP sensitivity benefited from amplifying more than one target.
Later, the saliva samples were also analyzed at NIB. The other laboratory has been advised to use its default methodology. NIB has been routinely using two technical replicates per RT-qPCR or RT-LAMP reaction. Both protocols, S-5/15 and LAMP-5/15, were used in a laboratory with established diagnostics in other fields and accredited according to ISO/IEC 17025 (NIB). Thus, the saliva samples were also tested in a way recommended for plant pathogenic microorganism detection, where more than one positive reaction indicates a positive result of the sample due to zero-tolerance (false-negative results can cause serious economic and social consequences).
Please, see attached file.

Reviewer 2 Report
The manuscript by Rajh et al is an interesting well-written study with sound research methodology and considerable number of samples for validation of an RNA extraction free method for COVID-19 fast detection. Few questions as below:
- The authors need to highlight if there is any potential effect of self-sampling on the assay?
- More details on the comparison of the current study method with reference no. 23 who used saliva and kind of similar buffer, but they extracted RND before the test in terms of differences in sensitivity and specificity
- As the authors indicated, testing of populations with low SARS-CoV-2 infection prevalence, sample pooling can be useful, based on this study, is it a recommendation to not exceed pooling of 5 samples? Please make it clear in the text to which LoD to go for pooling of these 5 samples (i.e. what ct value)
- The authors indicated that COVID-19 hospitalized samples from two clinics were not paired with NPS, so based on what the authors claimed that saliva is less sensitive in the late stages
- Line 323: “the period of three months” please correct to “a period of three months” unless there is a known character for these 3 months
- Line 371: the authors indicated “The saliva sampling method was an important step for diagnostic sensitivity of SARS-CoV-2 RNA amplification” add a reference
- Line authors indicated that “the net cost for the test ranges between 3-4 €” is this based on your calculations or the reference, also please add the current methodology cost to compare
- please indicate what TCEP stands for
- in tables S5, S6, S7. Why I see a difference in the number of total samples, this is confusing unless the authors justify
Author Response
The manuscript by Rajh et al is an interesting well-written study with sound research methodology and considerable number of samples for validation of an RNA extraction free method for COVID-19 fast detection. Few questions are below:
The authors need to highlight if there is any potential effect of self-sampling on the assay?
While developing the methodology, we identified several critical points in self-sampling: a clarity of self-sampling guidelines for individuals, sampling time, amount of collected saliva, and after-sampling procedure.
The critical points in self-sampling have been described in the Material and method, the Results, and the Discussion sections.
More details on the comparison of the current study method with reference no. 23 who used saliva and kind of similar buffer, but they extracted RND before the test in terms of differences in sensitivity and specificity
Howson et al. [27] determined that a combination of an ion-exchanger Chelex, Mucolyse, and heat treatment provides the highest sensitivity of RT-LAMP for viral RNA in saliva. However, they studied only a limited number of saliva samples; therefore, the diagnostic sensitivities were not determined. Nevertheless, we used their study as the backbone while developing our RNA stabilization buffer and have added the relevant information in the revised draft.
[27] Howson, E.L.A.; Kidd, S.P.; Armson, B.; Goring, A.; Sawyer, J.; Cassar, C.; Cross, D.; Lewis, T.; Hockey, J.; Rivers, S.; et al. Preliminary optimisation of a simplified sample preparation method to permit direct detection of SARS-CoV-2 within saliva samples using reverse-transcription loop-mediated isothermal amplification (RT-LAMP). J. Virol. Methods 2021, 289, 114048, doi:10.1016/j.jviromet.2020.114048.
As the authors indicated, testing of populations with low SARS-CoV-2 infection prevalence, sample pooling can be useful, based on this study, is it a recommendation to not exceed pooling of 5 samples? Please make it clear in the text to which LoD to go for pooling of these 5 samples (i.e. what ct value)
We have not tested the pooling of more than five samples, since with the pooled five samples, the Cq already increased, causing a decrease in sensitivity. The CDC has analyzed the pooling of 5, 10, and 20 samples (https://wwwnc.cdc.gov/eid/article/27/4/20-4200_article) and concluded that pooling should be considered only when the prevalence of infection is low. Exceeding a pool size of 5 can lead to low sensitivity and would be economically justified only with very low infection prevalence. In our case, the pool size increased the Cq, on average, to 4.6, which clearly shows that samples with Cq over 35 would not be detected. However, this problem can be circumvented by increasing the volume of the tested pooled samples. We have added a discussion on this topic in the revised draft.
The authors indicated that COVID-19 hospitalized samples from two clinics were not paired with NPS, so based on what the authors claimed that saliva is less sensitive in the late stages
All hospitalized patients were tested as SARS-CoV-2 positive using the NPS-RT-qPCR method. The test was performed at least 7 days before hospital admission. However, the saliva was collected approximately 10 days after NPS sampling. According to the literature, the viral load in saliva decreases in later stages of infection; therefore, we concluded that, due to the low viral load in saliva during a later stage of infection, the saliva specimen is not reliable for detecting SARS-CoV-2 infection.
In the revised manuscript, we have rephrased this text to clarify the above point.
Line 323: “the period of three months” please correct to “a period of three months” unless there is a known character for these 3 months
We have corrected the text as suggested.
Line 371: the authors indicated “The saliva sampling method was an important step for diagnostic sensitivity of SARS-CoV-2 RNA amplification” add a reference
The statement refers to our observation comparing oral cavity swabs with the passive drool saliva collection method. We have rephrased the sentence for clarity.
Line authors indicated that “the net cost for the test ranges between 3-4 €” is this based on your calculations or the reference, also please add the current methodology cost to compare
The prices for the RT-qPCR enzymes, primer–probe sets, chemicals, and laboratory consumables have changed during the six months; therefore, we decided not to include the net price figures in the text. We have rephrased the Conclusions as per Reviewer 3’s suggestion.
Please indicate what TCEP stands for
TCEP stands for tris(2-carboxyethyl)phosphine. The abbreviation has been added at the first instance.
In tables S5, S6, S7. Why I see a difference in the number of total samples, this is confusing unless the authors justify
Thank you for pointing out the discrepancies.
Let us briefly clarify the content and numbers in the tables. Table S5 depicts RT-LAMP and RT-qPCR results of different sampling methods. However, for the nylon swab, four samples were damaged, and for passive drool, some negative samples were not analyzed. Table S6 summarizes RT-LAMP; one sample was lost while running the LAMP-5/15 protocol. Table S7 summarizes RT-qPCR; three and two samples were lost while running multiplexes and the S-5/15 protocol, respectively.
For transparency, we have included an Excel Spreadsheet with raw results as Supplementary Material.

Reviewer 3 Report
The work is relevant to the ongoing pandemic and good comparison between the qPCR and LAMP. However, given that the current problem is the delta strain, it might be worth mentioning if the clinical samples were delta patients or earlier, and whether delta could be detected - this would improve the relevance of the article.
There is a lack of LAMP pictures given the colorimetric aspect. Authors mentioned using smartphones as a relatively throwaway sentence, might want to look at DOI: 10.30943/2021/17032021
The conclusion is written in an odd format, might want to rewrite that part to be more specific.
It may be good to state how the primers were designed, e.g. software for the LAMP primers.
Author Response
Reviewer 3.
The work is relevant to the ongoing pandemic and good comparison between the qPCR and LAMP. However, given that the current problem is the delta strain, it might be worth mentioning if the clinical samples were delta patients or earlier, and whether delta could be detected - this would improve the relevance of the article.
The saliva/NPS samples were collected during December 1st, 2020, to April 1st, 2021. At that time, no SARS-CoV-2 Delta strain was detected in Slovenia. However, we reran the in silico analysis of mutation frequencies within the primer–probe sequences of SARS-CoV-2 for the period of September 7th to October 7th, 2021 (over 80% prevalence of Delta strain). The data are listed in Table S4. In the Discussion, a paragraph discussing the selection of primer–probe sets has been added.
There is a lack of LAMP pictures given the colorimetric aspect. Authors mentioned using smartphones as a relatively throwaway sentence, might want to look at DOI: 10.30943/2021/17032021
For detecting the amplification efficiency of RT-LAMP, we used a real-time fluorescence detection of fluorescent dye Syto9. This procedure provides accurate results and enables the determination of the melting temperature and discrimination of nonspecific amplicons. For comparing all three RT-LAMP detection strategies, we have added Fig S2, which describes end-point colorimetric and fluorescence detection and real-time fluorescence detection.
Figure S2. (A) Colorimetric and (B) fluorescence end-point detection of RT-LAMP amplicons. (C) Real-time fluorescence detection of RT-LAMP amplicon and melting curve. Protocol LAMP-2/20 with N2 primers set was used to amplify SARS-CoV-2 mRNA.
The conclusion is written in an odd format, might want to rewrite that part to be more specific.
The text in this section has been thoroughly rewritten. We hope that the new arrangement conveys the findings more clearly.
It may be good to state how the primers were designed, e.g. software for the LAMP primers.
The RT-LAMP primers have been selected based on publications (12,20,21). Zhang et al. [20] used online software Primer Explorer V5 (https://primerexplorer.jp/e/) to design N2 and E1 primer sets. Kellner et al. [12] tested the amplification efficiency and specificity of six different primer sets (DETECTR N-gene, NEB E1, As1, NEB N-gene, NEB N2, and DETECTR RNaseP POP7), and based on these results, we selected NEB N2 and E1 primer sets for amplifying RBA from the saliva samples.
A description has been included in the Results section (2.2.).
[12] Kellner, M.J.; Ross, J.J.; Schnabl, J.; Dekens, M.P.S.; Heinen, R.; Grishkovskaya, I.; Bauer, B.; Stadlmann, J.; Menéndez-Arias, L.; Fritsche-Polanz, R.; et al. A rapid, highly sensitive and open-access SARS-CoV-2 detection assay for laboratory and home testing. bioRxiv (2020) 2020.06.23.166397 2020, doi:https://doi.org/10.1101/2020.06.23.166397.
[20] Zhang, Y.; Ren, G.; Buss, J.; Barry, A.J.; Patton, G.C.; Tanner, N.A. Enhancing colorimetric loop-mediated isothermal amplification speed and sensitivity with guanidine chloride. Biotechniques 2020, 69, 178–185, doi:10.2144/btn-2020-0078.
[21] Dong, Y.; Wu, X.; Li, S.; Lu, R.; Li, Y.; Wan, Z.; Qin, J.; Yu, G.; Jin, X.; Zhang, C. Comparative evaluation of 19 reverse transcription loop-mediated isothermal amplification assays for detection of SARS-CoV-2. Sci. Rep. 2021, 11, 2936, doi:10.1038/s41598-020-80314-0.
Please, see attached file.

Reviewer 4 Report
The manuscript by Rajh and colleague describes a new extraction-free one step RT-qPCR and RT-LAMP SARS-CoV-2 test that could be potentially used for mass testing using saliva samples. This article explores several interesting points, which include a comparison between RT-LAMP and RT-qPCR approaches, the use of appropriate paired NPS samples and the use of different saliva collection methods.
Although this is an interesting study with potential practical applications, I am quite perplexed by several choices made by the authors, and I would suggest them to improve the design of the test in order to provide more reliable data. One of the major issues I find in this work is the continuous use of the term “multiplex PCR” to indicate a PCR that actually used two different sets of primers targeting the very same gene (N1 and N2). Although this could be nearly considered as a sort of a technical replicate, this choice was quite unusual, as the overwhelming majority of SARS-CoV-2 commercial kits actually use 3 different sets of primers, with each of these targeting a different gene (e.g. S, N, E, ORF1ab, etc.).
Overall, considering the sensitivity of the topic addressed by the authors, I do not think that the methodology proposed here is mature for enough for publication. The use of saliva samples is, in perspective, very interesting and the use of a RNA stabilizing solution and the information concerning the performance of different saliva collection methods is extremely interesting. Still, the manuscript suffers in my opinion from a lack of clarity in the structure and the choice of using a non-standard set of primers for target amplification really damaged the reliability of the results.
I am quite sorry that my evaluation could not be more positive at this time, but at the same time I feel like this topic is really important in light of the public health consequences the use of new testing kits might have, and therefore I firmly believe that the any possible issue should be appropriately addressed before the publication of a new protocol of this kind.
Line 74: correct “bypas”
Line 138: N1 and N2 are not two different genes; theseaximal care should be put in s refer to two different regions of the N gene
Lines 141-142: this would need some additional discussion. “Rapidly evolving” does not mean much by itself: please provide some actual estimates (e.g. about 2 substitution per month, 25 per year, 8x10-4 subs/site/year, etc.). These can be easily extracted from GISAID. Also, one might note that such changes are not randomly distributed across the genome, as they disproportionally occur in the S gene, and precisely in the S1 region, which can be considered a mutational hotspot.
Line 145: its is not clear how such 28,555 sequences were selected. Was this due to random sampling? Or were only sequences deposited in a defined interval of time analyzed. Also, while talking about “currently circulating viral subtypes” the authors might want to update the text by stating that currently delta dominates the scenario, to the point that only the monitoring of delta-related AY.n lineages might be needed at this time.
Lines 269-272: this is quite confusing. N1 and N2 primers actually amplify the very same gene, targeting different regions. Hence, the claim made here by the authors does not seem to apply. Also, the rationale for certifying a sample as positive based on the amplification of just a single target is unclear. Such an approach would be expected to include in the pool of positive samples more samples with borderline Ct values, and possibly more false positives. This seems also to be contrary to usual laboratory practice since, to the best of my knowledge, uncertain tests with mixed positive/negative results for different targets should be repeated or considered as undetermined.
Figure 5: for multiplex PCR, do the Cq values displayed here represent the average Cq obtained from the different targets? How were mixed results (i.e. positive/negative) managed in this case, considering that the authors stated that all samples were the amplification of at least one target was successful were considered positive?
Line 494: why were different Ct thresholds used for the assessment of positivity with the N1 and N2 primer sets.
Section 5.5: the strategy used for the analysis of NPS was strikingly different and, in my opinion, much more appropriate than the one chosen by the authors. The use of two different target genes (E and RdRp) here configures a real multiplex PCR. I really do not see any valid reason why this approach was replaced, in the analysis of saliva samples by RT-qPCR, by a much less reliable approach, which either used a singleplex PCR or a pseudo-multiplex PCR (with two sets of primers targeting the same gene). Also, why was a different Ct (i.e. 38) used in this case?
Author Response
The manuscript by Rajh and colleague describes a new extraction-free one step RT-qPCR and RT-LAMP SARS-CoV-2 test that could be potentially used for mass testing using saliva samples. This article explores several interesting points, which include a comparison between RT-LAMP and RT-qPCR approaches, the use of appropriate paired NPS samples and the use of different saliva collection methods.
Although this is an interesting study with potential practical applications, I am quite perplexed by several choices made by the authors, and I would suggest them to improve the design of the test in order to provide more reliable data. One of the major issues I find in this work is the continuous use of the term “multiplex PCR” to indicate a PCR that actually used two different sets of primers targeting the very same gene (N1 and N2). Although this could be nearly considered as a sort of a technical replicate, this choice was quite unusual, as the overwhelming majority of SARS-CoV-2 commercial kits actually use 3 different sets of primers, with each of these targeting a different gene (e.g. S, N, E, ORF1ab, etc.).
Overall, considering the sensitivity of the topic addressed by the authors, I do not think that the methodology proposed here is mature enough for publication. The use of saliva samples is, in perspective, very interesting and the use of a RNA stabilizing solution and the information concerning the performance of different saliva collection methods is extremely interesting. Still, the manuscript suffers in my opinion from a lack of clarity in the structure and the choice of using a non-standard set of primers for target amplification really damaged the reliability of the results.
I am quite sorry that my evaluation could not be more positive at this time, but at the same time I feel like this topic is really important in light of the public health consequences the use of new testing kits might have, and therefore I firmly believe that the any possible issue should be appropriately addressed before the publication of a new protocol of this kind.
We appreciate the reviewer’s concerns and offer the following explanation. The N1 and N2 primer–probe sets amplify two regions within the same N gene. We have corrected this error throughout the text.
Nevertheless, amplifying two or more nonidentical targets within a reaction should be considered multiplex. Just for demonstration, the N gene N1 and N2 primer–probe sets target different (relatively distant) regions of the N gene, corresponding to the fragments from 28,287-28,358 (N1) and 29,164-29,230 (N2). N2 is (only) two-fold closer to N1 compared to, for example, gene E, (as illustrated in Fig 1 in https://doi.org/10.1016/j.jare.2020.08.002). This also implies that the detection of one N region and E gene (or any other gene in SARS-CoV-2 RNA) is a sort of technical replicate.
The viral targets for RT-LAMP and RT-qPCR amplification were carefully selected based on the literature and guidelines offered by CDC. Most laboratories and IVD kits use the N1 and N2 targets with their assays. Furthermore, several published papers have supported the selection of N1 and N2 targets as multiplex for detecting SARS-CoV-2 virus.
Links to literature:
https://covid-19-diagnostics.jrc.ec.europa.eu/devices/5?device_id=&manufacturer=&text_name=&marking=&method=&rapid_diag=&target_type=5&field-1=Assay%20category&value-1=1&field-2=Assay%20category&value-2=2&search_method=OR#form_content;
https://www.fda.gov/media/142539/download;
https://www.fda.gov/media/134922/download;
https://www.cdc.gov/coronavirus/2019-ncov/downloads/rt-pcr-panel-primer-probes.pdf
https://www.bd.com/en-us/offerings/capabilities/molecular-diagnostics/molecular-tests/biogx-sars-cov-2-reagents;
https://www.optimedical.com/en/products-and-services/kits/opti-sars-cov-2-rt-pcr-test-kit/;
https://doi.org/10.1016/j.jcv.2020.104499,
https://doi.org/10.1016/j.ijid.2020.10.047;
https://doi.org/10.1016/j.medj.2020.12.010
We decided to use N1 and N2 targets since the protocol was established by CDC, and is, according to CDC and FDA, adequate for testing and monitoring of SARS-CoV-2 (e.g., NIB participates in the EU framework for monitoring of wastewater/sewage for SARS-CoV-2; COVID-19 Tracking (sledilnik.org) for Slovenia). Furthermore, before starting the RT-qPCR analysis for SARS-CoV-2, we evaluated the performance of assays targeting the E1 gene and RdRp gene and N1 and N2 regions of the N gene. E1 and RdRP showed lower sensitivity and were, therefore, replaced with N1 and N2 targets of the N gene. From the in silico analysis of the mutation frequency, a primer RdRP_F target region has a very high rate of mutation (Table S4); therefore, the primer sequence should be adjusted.
Nevertheless, to reassure the reviewer, we have simultaneously performed multiplexes RT-qPCR (M-2.5/10) using N1/N2 and E1/N2 primer sets on the remaining saliva samples. As expected, similar results were obtained regardless of the primer–probe set selected. The results have been included as a supplemental figure (Fig S3) and the relevant information has been added in the revised draft.
Figure S3. The heat map of matched Cq values. Pooled Cq values with depicted average value as a red line (below).
Line 74: correct “bypas”
Corrected
Line 138: N1 and N2 are not two different genes; theseaximal care should be put in s refer to two different regions of the N gene
Corrected
Lines 141-142: this would need some additional discussion. “Rapidly evolving” does not mean much by itself: please provide some actual estimates (e.g. about 2 substitution per month, 25 per year, 8x10-4 subs/site/year, etc.). These can be easily extracted from GISAID. Also, one might note that such changes are not randomly distributed across the genome, as they disproportionally occur in the S gene, and precisely in the S1 region, which can be considered a mutational hotspot.
The estimated mutation rate based on the two publications ranges from 1.3 to 3.3 ×10-3 per site per year [23,24]; however, some regions, like S1, mutate faster. The target regions selected for the RT-qPCR and RT-LAMP primer–probe sets were deliberately selected in low-mutation-rate sections.
The Result section (2.2.) has been modified to include these data and references.
[23] Li, X.; Zai, J.; Zhao, Q.; Nie, Q.; Li, Y.; Foley, B.T.; Chaillon, A. Evolutionary history, potential intermediate animal host, and cross‐species analyses of SARS‐CoV‐2. J. Med. Virol. 2020, 92, 602–611, doi:10.1002/jmv.25731.
[24] Chaw, S.-M.; Tai, J.-H.; Chen, S.-L.; Hsieh, C.-H.; Chang, S.-Y.; Yeh, S.-H.; Yang, W.-S.; Chen, P.-J.; Wang, H.-Y. The origin and underlying driving forces of the SARS-CoV-2 outbreak. J. Biomed. Sci. 2020, 27, 73, doi:10.1186/s12929-020-00665-8.
Line 145: its is not clear how such 28,555 sequences were selected. Was this due to random sampling? Or were only sequences deposited in a defined interval of time analyzed. Also, while talking about “currently circulating viral subtypes” the authors might want to update the text by stating that currently, delta dominates the scenario, to the point that only the monitoring of delta-related AY.n lineages might be needed at this time.
The analysis of mutation frequency within the primer–probe sets was performed on all SARS-CoV-2 sequences deposited during March 21st to April 20th, 2020, from EU (very low prevalence of Delta strain). We reran the analysis on SARS-CoV-2 sequences deposited from EU from September 7th to October 7th, 2021 (very high prevalence of Delta strain). The results of the in silico analysis of the mutation frequencies are presented in Table S4 and discussed.
Lines 269-272: this is quite confusing. N1 and N2 primers actually amplify the very same gene, targeting different regions. Hence, the claim made here by the authors does not seem to apply. Also, the rationale for certifying a sample as positive based on the amplification of just a single target is unclear. Such an approach would be expected to include in the pool of positive samples more samples with borderline Ct values, and possibly more false positives. This seems also to be contrary to usual laboratory practice since, to the best of my knowledge, uncertain tests with mixed positive/negative results for different targets should be repeated or considered as undetermined.
We have amended the conflicting sentences, stressing that two regions within the same gene were selected as targets for multiplex RT-qPCR.
We agree that for diagnostics, the reliability of the result increases with the number of positive hits. Since the methodology was developed primarily as an economically sound mass testing strategy, the “so-called” mixed positive/negative results were also declared as positive.
We agree that for diagnostic purposes, the mixed positive/negative results should be repeated. However, for fast diagnosis, where each suspicious result could mean infection, we decided to interpret the results with only one positive reaction as positive. These persons would, in real life, be sent to official testing.
Figure 5: for multiplex PCR, do the Cq values displayed here represent the average Cq obtained from the different targets? How were mixed results (i.e. positive/negative) managed in this case, considering that the authors stated that all samples were the amplification of at least one target was successful were considered positive?
To avoid confusion, Figure 5D–F (multiplex RT-qPCR and S-5/15) was amended to depict the Cq values of individual N2 and N1 primer–probe sets. In addition, an Excel Spreadsheet with all results has been presented as the supplementary Material.
Line 494: why were different Ct thresholds used for the assessment of positivity with the N1 and N2 primer sets.
The Cq value of the results was set based on the evaluation of the protocols on spiked INSTAND material in saliva and analysis of real samples. The table above presents the results of qPCR and dPCR on INSTAND material spiked in saliva and tested directly (no RNA extraction). To determine the highest Cq value, which is still positive, we used the data from two individual dilution series, tested on two separate days. In both repetitions, we determined a very similar cut-off Cq value, which indicated high repeatability of the assays. However, the cut-off value was different between the assays N1 and N2, which is a normal and well-known phenomenon, as two different assays cannot be expected to have the same Cq values.
Section 5.5: the strategy used for the analysis of NPS was strikingly different and, in my opinion, much more appropriate than the one chosen by the authors. The use of two different target genes (E and RdRp) here configures a real multiplex PCR. I really do not see any valid reason why this approach was replaced, in the analysis of saliva samples by RT-qPCR, by a much less reliable approach, which either used a singleplex PCR or a pseudo-multiplex PCR (with two sets of primers targeting the same gene). Also, why was a different Ct (i.e. 38) used in this case?
The NPS samples have been analyzed as part of the National Health Care Scheme for SARS-CoV-2 detection. The RT-qPCR using E1 and RdRP as targets have been conducted by an outside laboratory. Furthermore, the experimental conditions are described by the manufacturer (also Cq).
In contrast, the one-step saliva protocol was developed simultaneously in the research institutes as a backup protocol, diverting from their methodology. Therefore, the selection of primers, reagents, and sampling procedures differ between laboratories and official diagnostic centers.
However, the targets for amplification and primer/probe selection were not random. The N1 and N2 targets were selected based on the CDC recommendation on the RNA loci with minimal mutation frequency, which could not be said for the RdRP region (Table S4). The literature has corroborated the appropriateness of the target selection. The rationale behind the selection of the primer–probe sets for RT-qPCR was included in the Material and methods and Results sections. Furthermore, we have run the comparison of N1/N2 and E1/N2 multiplexes on a limited number of samples. The results confirmed that primer selection in a multiplex has a minor impact on the detection of SARS-CoV-2. The results have been presented as supplementary data.
We believe that the present study is a very important and needed report showing the robustness of SARS-CoV-2 testing protocols, differently performed in differently organized laboratories. The true significance of the present study, in addition to the development of fast sampling and testing protocol, lies in the evaluation of these protocols and the presentation of the impact they might have on the final decisions. The authors’ take-home message from the manuscript is also to encourage diagnostic laboratories to conduct similar robustness studies and evaluate their performance.
Please, see attached file.

Round 2
Reviewer 4 Report
I appreciate the explanations provided by the authors, in particular for what concerns the choice of the primer sets, and the additional efforts made to improve the manuscript. While I still maintain some reservations about a few points, considering the assessment made by the other reviewers, and the take home message of this work, I believe that the manuscript may deserve publication, pending some additional modifications.
Mutation rate: the two cited references are quite outdated, as they refer to the early stages of the pandemics (and in one case, I believe, they concern the rate observed at the 3rd codon only). With millions viral genomes sequenced, we are now able to provide a much more precise and reliable estimate of the substitution rate (regardless of codon position), which currently stands between 24 and 25 subs/year/genome (this can be easily used to calculate the rate per site per year). The authors might simply cite GISAID, using the most recent estimate which can be extracted from the global nextstrain build – just select “clock” from the side panel in the tree visualization page) As of Oct 21st, the estimate is 24,604 subs/genome/year.
“ We agree that for diagnostic purposes, the mixed positive/negative results should be repeated. However, for fast diagnosis, where each suspicious result could mean infection, we decided to interpret the results with only one positive reaction as positive. These persons would, in real life, be sent to official testing.”
Thanks for the explanation. This sounds reasonable indeed. I would just suggest to add a few sentences in the discussion to explain this caveat.
“However, the cut-off value was different between the assays N1 and N2, which is a normal and well-known phenomenon, as two different assays cannot be expected to have the same Cq values.”
I don’t find this explanation convincing. While it is definitely true that the amplification of two different targets in qRT-PCR is not expected to result in the very same Ct due to different amplification efficiency, any upper threshold for detection will be still arbitrarily set. These are usually set regardless of the primers and probes used and of the amplification of a given target, but they are rather defined with the aim to reduce avoid the chances of false positive detection and other unwanted results which may arise from excessive cycling and reagent (e.g. dNTPs) depletion. For example, no commercial SARS-CoV-2 detection kit for qRT-PCR used different thresholds for different targets. Hence, I would strongly suggest the authors to apply a unique arbitrary threshold for all targets. I doubt this is going to change significantly their results, but I am pretty sure it would sound more reasonable than it currently sounds both to the readers and policy makers.
Author Response
I appreciate the explanations provided by the authors, in particular for what concerns the choice of the primer sets, and the additional efforts made to improve the manuscript. While I still maintain some reservations about a few points, considering the assessment made by the other reviewers, and the take home message of this work, I believe that the manuscript may deserve publication, pending some additional modifications.
1. Mutation rate: the two cited references are quite outdated, as they refer to the early stages of the pandemics (and in one case, I believe, they concern the rate observed at the 3rdcodon only). With millions viral genomes sequenced, we are now able to provide a much more precise and reliable estimate of the substitution rate (regardless of codon position), which currently stands between 24 and 25 subs/year/genome (this can be easily used to calculate the rate per site per year). The authors might simply cite GISAID, using the most recent estimate which can be extracted from the global nextstrain build – just select “clock” from the side panel in the tree visualization page) As of Oct 21st, the estimate is 24,604 subs/genome/year.
We have amended the text as Reviewer 4 suggested.
Revised text: The temporal resolution assumes a nucleotide substitution rate of 8 ×10-4 substitutions/site/year (approximately 24600 substitutions/genome/year; https://nextstrain.org/ncov/gisaid/global?l=clock) [23] with the mutations concentrated within hotspots, one of which is the S1 protein.
2. “ We agree that for diagnostic purposes, the mixed positive/negative results should be repeated. However, for fast diagnosis, where each suspicious result could mean infection, we decided to interpret the results with only one positive reaction as positive. These persons would, in real life, be sent to official testing.” Thanks for the explanation. This sounds reasonable indeed. I would just suggest adding a few sentences in the discussion to explain this caveat.
Thanks for the advice. This sounds reasonable indeed. We added a few sentences in the discussion to explain this caveat.
3. “However, the cut-off value was different between the assays N1 and N2, which is a normal and well-known phenomenon, as two different assays cannot be expected to have the same Cq values.”
I don’t find this explanation convincing. While it is definitely true that the amplification of two different targets in qRT-PCR is not expected to result in the very same Ct due to different amplification efficiency, any upper threshold for detection will be still arbitrarily set. These are usually set regardless of the primers and probes used and of the amplification of a given target, but they are rather defined with the aim to reduce avoid the chances of false positive detection and other unwanted results which may arise from excessive cycling and reagent (e.g. dNTPs) depletion. For example, no commercial SARS-CoV-2 detection kit for qRT-PCR used different thresholds for different targets. Hence, I would strongly suggest the authors to apply a unique arbitrary threshold for all targets. I doubt this is going to change significantly their results, but I am pretty sure it would sound more reasonable than it currently sounds both to the readers and policy makers
We understand the concern regarding the use of set thresholds from the readers' and policymakers' points of view. The S-5/15 protocol was developed and evaluated by NIB, which is an expert in the field of plant pathology, GMO detection, and metrology (designated as EURL, NRL, holder of national etalon). Based on extensive experience in the introduction of methods for the low-level presence of targets, a unique arbitrary threshold for all targets is not the optimal option. Setting the threshold should not be unique for all protocols as we have shown that results of quantification can vary thousands of fold when measuring viral genomes (https://doi.org/10.1016/j.ymeth.2021.03.016). Just recently, a publication showing results from the SARS-CoV-2 testing with qPCR, stating that the “Cq values should not be used for cut-off setting to stratify risk or guide analytical performance” (Dangers of using Cq to quantify nucleic acid in biological samples; a lesson from COVID-19 | Clinical Chemistry | Oxford Academic (oup.com)).
Consequently, NIB has set the Cq values following internal procedure (part of the protocol for method validation in the GMO detection laboratory accredited according to ISO/IEC 17025) described below.
The Cq values obtained during the repeatability runs of method validation were carefully checked. In the area close to the LoD, where some wells started yielding »undetermined« (negative) results, the range of the highest Cq values was observed (not considering out-layers). The highest Cq value was rounded to the half value above, and then 0.5 was added to this Cq value to account for the potential difference in the thresholds chosen between runs. For the N1 region of the N gene, we obtained the highest Cq value as 41.73, which was rounded to 42.0, and 0.5 was added, resulting in a cut-off value of 42.5. This cut-off value was also confirmed on the real saliva samples. In the case of the N2 region, we obtained extremely high Cq values in the repeatability study, which gave a cut-off value of 45.5 (when using the same logic as that used for N1). However, when analyzing the Cq values from the real saliva samples, we concluded that such high values normally occur in only one replicate and that in most of those samples, the N1 assay gave a negative result. To minimize the possible false-positive interpretation of the results, we decided to use the rounded average of the Cq value of the last positive sample from the repeatability study as the cut-off value for the N2 assay (43.5).
Just for illustration, a proportion of individuals with Cq > 38 was approximately 1 of 99 (S-2/20), 4 of 96 (N2 target M-5/20), 2 of 99 (N1 target M-2.5/10), 17 of 98 (N1 target S-5/15), and 26 of 98 (N2 target S-5/15).
We have added a brief description of setting the threshold Cq in section 5. Materials and methods and described the results in section 2.2 Detection of viral RNA at low concentrations.
Round 3
Reviewer 4 Report
Thank you for providing a revised version with satisfactory modifications. I have no further comments.